# OpenReview forum: "FlowNar: Scalable Streaming Narration for Long-Form Videos"
_ICLR.cc/2026/Conference — ICLR 2026 Conference Desk Rejected Submission_

### Official Review · Reviewer_jfH5 · 2025-10-23

**Soundness:** 3
**Presentation:** 3
**Contribution:** 3
**Rating:** 8
**Confidence:** 3

**Summary:**

This paper presents FlowNar, a framework designed to generate narrations for streaming videos. The key challenge addressed is that existing methods for video narration require memory and computation that grows linearly (or worse) with video length, making them impractical for long videos. FlowNar introduces two main contributions: (1) DCM, which removes detailed visual information from past video segments after generating each narration, and (2) CLAM, a novel module that compresses historical visual information into a fixed-size memory bank. The authors also propose evaluation protocol where the model generates narrations based on its own previous outputs rather than ground truth. Experiments on three egocentric video datasets (Ego4D, EgoExo4D, and EpicKitchens100) demonstrate that FlowNar can process videos 10× longer than baseline methods while achieving 3× higher throughput and improved narration quality.

**Strengths:**

- The CLAM module is an interesting adaptation of linear attention mechanisms for streaming video. The design maintains constant memory and per-step computation, also the combination of DCM and CLAM provides a complete solution to the scalability challenge.
- The paper introduces a realistic autoregressive evaluation protocol that better reflects deployment conditions. The "first-align-then-evaluate" procedure is a thoughtful solution to the challenge of evaluating predictions with different temporal boundaries than ground truth.
- The paper includes extensive ablations that validate each component's contribution. The analyses of visual history strategies, memory designs, and trigger mechanisms provide good insights into why the approach works.

**Weaknesses:**

- The dynamic threshold mechanism requires setting several hyperparameters ($\theta$, $\theta_\mathrm{low}$, refractory period). Table 11 in the appendix shows that performance varies with threshold values. The paper does not provide clear guidance on how to set these parameters for new domains or video types, which could limit the method's generalizability.
- All three datasets focus on egocentric videos, primarily of kitchen and daily activities. The evaluation does not include other important video types such as sports, movies, educational content, or surveillance footage. This raises questions about how well the approach generalizes to different video genres.
- I'm not convinced about $\theta_\mathrm{low}$. Consider a scenario with dense information content (e.g., a streaming educational lecture with continuous new information). After each narration, the method applies $\theta_\mathrm{low}$, which discourages triggering. Could this cause the model to miss important content that should trigger the next narration? How does the method handle videos with consistently high information density?

**Questions:**

- Does the "first-align-then-evaluate" scheme specifically favor FlowNar over the baselines? Could this evaluation approach be inherently biased toward methods that generate narrations at specific temporal patterns?
- The CLAM module uses a fixed M×D memory regardless of video length. Consider two scenarios: 3 previous frames vs 30 previous frames. Does the 30-frame scenario suffer from more information loss since memory size is fixed? Or is this similar to language models where the last token contains information from all previous layers, so there's no information loss?
- For the fluency and LM-correctness metrics, both FlowNar and baselines use the same fixed LLM (Llama-3-8B or 1B). Since these metrics depend on the LM used and all methods use the same one, does comparing these metrics provide meaningful differentiation? Could you clarify what these metrics are actually measuring?
- In Table 3, using no past frames outperforms using all past frames by a large margin (30.40 vs 28.04 CIDEr), which is a bit surprising. Can the authors explain why having no historical visual information would be way better than having complete history?
- In lines 428-429, the authors write that "providing past visual context generally improves narration quality for the oracle protocol (cf. row 1 vs. row 2/3)." However, looking at Table 4, I don't see a clear pattern supporting this conclusion. The differences in PPL and TimeDiff are very small (2.122 vs 2.115 vs 2.118). Could the authors clarify this claim?
- The paper compares against Videollm-online, Videollm-mod, and LION-FS. Several other recent streaming video methods are mentioned (ProVideLLM, Dispider, LiveCC) but not experimentally compared. Could the authors clarify why these methods were not included in the comparison? Is it due to code availability, or fundamental differences in task formulation that make direct comparison difficult?
- The paper acknowledges that training time is increased due to longer sequences and less optimized attention kernels, but does not provide concrete numbers or comparison. Providing more information and experimental statistics will be helpful.

---

> ### Author Response · Authors · 2025-11-22
>
> We thank the reviewer for the positive assessment of our "thoughtful" evaluation protocol and for finding our extensive ablations valuable. We appreciate your detailed questions regarding hyperparameters and metrics, which have helped clarify key aspects of our work.
>
> 1. **Clear guidance on setting parameters / dense information content**: These hyperparameters primarily control narration frequency (temporal alignment) rather than narration quality, as shown in Table 11. In our work, we set two thresholds, one high (around 0.8) and one low (around 0.5), to trigger timely narrations while avoiding excessive repetitions, which would significantly increase latency. This setup worked well for all datasets. Only the refractory period is domain-dependent and as a rule of thumb it can be set as the average training segment duration, which we did in our experiments. For videos with consistently high information density, a shorter refractory period that matches the higher event frequency would be appropriate. We kindly note that while narration is discouraged during the refractory period, the final decision is still determined by the model (via $p_\texttt{[SKIP]}$). Even if a narration is deferred during this window, the visual frames are still cached and are incorporated at the next narration step, thereby reducing the risk of missing important content.
>
> 2. **Other video types**: We follow prior narration work by evaluating on Ego4D and EgoExo4D, and extend the evaluation to EpicKitchens100, which contains longer-duration videos. A primary reason for selecting these datasets is that they provide high-quality, fine-grained narrations that are necessary for training and evaluating streaming narration models. We agree that evaluating across a wider range of video types would be valuable, but the three datasets already cover challenging camera motion and very diverse daily activities in indoor and outdoor environments. Since our approach does not make any assumptions about egocentric videos, it can be applied to other video genres as well, as long as there is enough training data.
>
> 3. **Does "first-align-then-evaluate" scheme favor FlowNar**: The ``first-align-then-evaluate'' protocol is a generic, model-agnostic strategy for evaluating self-conditioned narrations. It does not favor FlowNar over the baselines.
>
> 4. **Fixed memory regardless of video length**: CLAM's recurrent state update follows the principles of linear attention models, where the current state selectively aggregates information from all past history. Because the state has a fixed size, information loss for very long histories is inevitable compared to full self-attention. To mitigate this, our model uses two complementary information sources: (1) the compressed historical information from CLAM, and (2) the uncompressed, detailed frame information within the current video segment. Empirically, this combination provides strong performance.
>
> 5. **Fluency and LM-correctness metrics**: We kindly note that the LLM is finetuned with LoRA for all methods. The fluency and LM-correctness metrics are not computed by an external "scoring LLM". Instead, they measure how well each model’s own language head predicts its generated narration tokens. These metrics have been proposed by Videollm-online for the oracle evaluation protocol and assess the internal linguistic consistency and correctness of each fine-tuned model.
>
> 6. **Table 3**: We attribute the lower performance of using "all past frames" to two factors: (1) Error Propagation: In the autoregressive setting, past segments are defined by previous predictions. If the model drifts, the visual history becomes misaligned. Including this misaligned history reinforces errors, whereas using "no past frames" acts as a visual reset. (2) Over-reliance on Historical Context: During training, the model receives ground-truth aligned history and may overfit to these "perfect" past cues, learning to rely on them as shortcuts even when they are less relevant than the current frame. During inference, when the history contains accumulated errors, this learned over-reliance causes the model to attend to noisy or irrelevant past frames, distracting it from the current visual input.
>
> 7. **Table 4**: We have rephrased the sentence in the revised paper. There is an improvement of PPL in Table 4 (row 2/3 compared to row 1), but the differences are smaller compared to the gain by CLAM (row 4).

---

> > ### Author Response · Authors · 2025-11-22
> >
> > 8. **Comparison to other streaming video methods**: These methods address tasks that differ from our online video narration setting. Specifically, ProVideLLM focuses on action anticipation and recognition, rather than narration generation. Dispider focuses on streaming video QA and interaction, and LiveCC targets real-time commentary of sports videos conditioned on streaming ASR transcripts. Their visual/language backbones, training objectives, and data pipelines differ substantially from ours, a comparison is thus not possible.
> >
> > 9. **Training time**: The total training time of the 1B-parameter model on Ego4D is 67 GPU-hours on H100s. Videollm-online requires for the same hardware 36 GPU-hours of training.

---

### Official Review · Reviewer_VDde · 2025-10-30

**Soundness:** 3
**Presentation:** 3
**Contribution:** 3
**Rating:** 6
**Confidence:** 4

**Summary:**

The paper proposes FLOWNAR, a novel framework for scalable streaming video narration using large multimodal models (LMMs). It addresses the core challenge that existing online video narration methods (e.g., Videollm-online) scale linearly in computational and memory cost with video length, limiting their use for long-form streaming scenarios.

**Strengths:**

1. FLOWNAR combines dynamic pruning with learned visual compression, achieving both constant memory and stable narration quality.
2. Extensive tests on three long-form datasets, with both oracle and autoregressive protocols.
3. Comprehensive analyses of memory design, triggering strategy, and DCM effects.
4. The writing is concise and technically precise; mathematical notation is consistent and readable.

**Weaknesses:**

1. While CLAM’s design is intuitive, there is limited theoretical analysis of its representational capacity or stability compared to standard linear attention mechanisms.
2. Although major baselines (Videollm-online, Videollm-mod) are included, additional recent streaming methods (e.g., Dispider, ProVideLLM) are only cited but not empirically compared.
3. While qualitative examples are shown (Fig. 5), more failure analysis or long-horizon error accumulation visualization would strengthen interpretability.

**Questions:**

1. How would CLAM compare to recurrent state-space models (e.g., Mamba or RetNet) for streaming visual summarization?
2. Have the authors evaluated FLOWNAR on non-egocentric datasets (e.g., HowTo100M or ActivityNet) to test domain transferability?
3. Does FLOWNAR exhibit “drift” in narration over multi-hour streams, and how does DCM frequency impact long-term consistency?

---

> ### Author Response · Authors · 2025-11-22
>
> We thank the reviewer for acknowledging the "concise and technically precise" writing and for highlighting our comprehensive analysis of memory design and triggering strategies. We appreciate your insightful questions regarding theoretical grounding and state-space models.
>
> 1. **Theoretical analysis of its representational capacity**: We kindly note that CLAM's recurrent state update follows the standard linear attention formulations, whose representational capacity scales linearly with the model dimension [1]. We have integrated this theoretical discussion into Section 3.3 of the revised paper.
>
> [1] Zhong, Shu, et al.``Understanding Transformer from the Perspective of Associative Memory.'' arXiv preprint arXiv:2505.19488 (2025).
>
> 2. **Comparison to recurrent state-space models**: Regarding comparison with Mamba or RetNet, we would like to point out these models are designed as language-modeling architectures with self-attention-like updates, rather than cross-modal visual summarization modules. Integrating Mamba into our framework would require substantial changes to the architecture (e.g., modifying the SSM kernels to support cross-attention-style readout). RetNet requires fewer modifications. We therefore refactored a RetNet-style update and included it in our ablations on Ego4D (Table 6):
>
> | Method        | PPL ↓     | TimeDiff ↓ | Fluency ↑ | LM-Corr. ↑ |
> | ------------- | --------- | ---------- | --------- | ---------- |
> | recent        | 2.115     | 2.257      | 45.0%     | 51.7%      |
> | K-Means       | 2.114     | 2.248      | 45.0%     | 51.7%      |
> | MovieChat     | 2.105     | 2.265      | 45.1%     | 51.9%      |
> | TokenMLP      | 2.127     | 2.274      | 44.6%     | 51.3%      |
> | Refac. RetNet | 2.092     | 2.239      | 45.3%     | 52.1%      |
> | **CLAM**      | **2.086** | **2.237**  | **45.4%** | **52.2%**  |
>
> CLAM also performs better than the refactored RetNet-style update, which we attribute to CLAM's learnable gating compared with RetNet's manually designed fixed scalar decay.
>
> 3. **Comparison to additional streaming methods**: These methods address tasks that differ from our online video narration setting. Specifically, ProVideLLM focuses on action anticipation and recognition, rather than narration generation. Dispider focuses on streaming video QA and interaction, and LiveCC targets real-time commentary of sports videos conditioned on streaming ASR transcripts. Their visual/language backbones, training objectives, and data pipelines differ substantially from ours, a comparison is thus not possible.
>
> 4. **Non-egocentric dataset**: We apply our Ego4D-trained model directly to ActivityNet without any finetuning and provide several qualitative examples (Fig. 6 and Fig. 13 of the revised paper). Since it is trained on egocentric videos, the narration is always from the first-person perspective while the narrations in ActivityNet are from the third-person perspective (e.g. *A women paints the lower part of a gate* vs. *You paint the gate*). Despite the different narration style and not using any training data from exocentric-view, FlowNar is able to correctly identify key activities in many cases (e.g., painting, playing a game).
>
> 5. **Drift and failure cases**: We study the effect of error accumulation by evaluating on EK100 under the autoregressive protocol and grouping videos into four duration ranges:
>
> | Duration (min) | Prec. ↑   | Rec. ↑    | F1 ↑      | C ↑       | M ↑       | R ↑       |
> | -------------- | --------- | --------- | --------- | --------- | --------- | --------- |
> | ≤ 2            | **57.05** | **32.58** | **40.21** | **77.47** | **17.07** | **49.21** |
> | 2–4            | 48.51     | 23.14     | 29.01     | 54.99     | 15.98     | 46.89     |
> | 4–8            | 44.07     | 17.80     | 23.15     | 59.60     | 15.96     | 47.05     |
> | ≥ 8            | 41.41     | 12.79     | 16.93     | 37.94     | 14.13     | 45.46     |
>
> We observe a gradual degradation in both temporal alignment and narration quality as video length increases, which reflects the expected *drift* phenomenon in long-horizon autoregressive narration. This drift is caused by feeding back the previous generated narration. In the oracle history protocol (see response to Reviewer YeT5), this drift is not observed. We will discuss this limitation.
>
>  Additionally, we have included a qualitative failure analysis in Appendix (Sec. A.4, Fig. 14). We observe that errors typically manifest as fine-grained object misclassifications (e.g., predicting contextually plausible objects like a *knife* instead of the active *spatula*). We attribute this to the spatial compression ($1\text{ CLS} + 3\times3$ pooled tokens), consistent with prior streaming baselines, which is required to maintain high streaming throughput. While this design ensures real-time performance, it limits precise object grounding by oversmoothing fine-grained visual details.

---

> > ### Author Response · Authors · 2025-11-22
> >
> > 6. **DCM frequency**: Regarding the impact of narration frequency (DCM frequency), Table 11 in the supplementary material shows that varying the trigger thresholds affects temporal alignment more strongly than narration quality. That is, narration frequency predominantly determines when narrations are emitted, while the content quality remains relatively stable across a reasonable range of the parameter.

---

> > > ### Comment · Reviewer_VDde · 2025-11-27
> > > **Response to rebuttal**
> > >
> > > Thank you for the detailed response. I still have some questions. To the best of my knowledge, there are many general-purpose video captioning works, such as:
> > > ```
> > > [1]"Autoad: Movie description in context." Proceedings of the IEEE/CVF Conference on Computer Vision and Pattern Recognition. 2023.
> > > [2]"Autoad ii: The sequel-who, when, and what in movie audio description." Proceedings of the IEEE/CVF International Conference on Computer Vision. 2023.
> > > [3]"Autoad iii: The prequel-back to the pixels." Proceedings of the IEEE/CVF Conference on Computer Vision and Pattern Recognition. 2024.
> > > [4]"Autoad-zero: A training-free framework for zero-shot audio description." Proceedings of the Asian Conference on Computer Vision. 2024.
> > > [5]"MMAD: Multi-modal movie audio description." Proceedings of the 2024 Joint International Conference on Computational Linguistics, Language Resources and Evaluation (LREC-COLING 2024). 2024.
> > > [6]"Osprey: Pixel understanding with visual instruction tuning." Proceedings of the IEEE/CVF Conference on Computer Vision and Pattern Recognition. 2024.
> > > [7]"Videorefer suite: Advancing spatial-temporal object understanding with video llm." Proceedings of the Computer Vision and Pattern Recognition Conference. 2025.
> > > ```
> > > As well as recent VLM-based video captioning methods, such as the QwenVL series.
> > >
> > > I am curious about the performance of these general video captioning methods on egocentric datasets. Recently, I directly used Qwen2.5VL/Qwen3VL to generate video captions for the Ego4D dataset, and the results seemed very good. Do you have any experimental results regarding these general methods? If experiments are not feasible, these methods should be considered in the related worl section, as the video captioning field is already quite mature.

---

> > > > ### Author Response · Authors · 2025-11-27
> > > >
> > > > We thank the reviewer for the comprehensive list of recent general-purpose video captioning and audio description works and appreciate the reviewer sharing their fruitful experience with Qwen-VL. We agree that the AutoAD series and foundation models like Qwen-VL represent the state-of-the-art in movie and general video understanding.
> > > >
> > > > We have analyzed these works and will include them in our Related Work section. However, we wish to clarify why a direct experimental comparison is not feasible due to fundamental differences in task formulation and experimental control:
> > > >
> > > > - **Audio Description (AD) vs. Streaming Narration**: There are two fundamental differences between the AutoAD series and FlowNar:
> > > >   1. Dependency on Non-Visual Context: The AutoAD series is explicitly designed for Movie Audio Description. These methods typically rely on current visual frames combined with movie subtitles and previous description history (e.g., see Section 3.2 in [1]) to resolve plot-heavy context. Their goal is to provide descriptions complementary to the soundtrack/dialogue. In contrast, FlowNar aims to describe human activity events based solely on visual information, making it applicable to unscripted scenarios where no subtitles or soundtrack exist.
> > > >   2. Clip-level vs. Frame-level: AutoAD methods operate at the clip level, where video segments are typically pre-defined based on ground-truth AD annotations (e.g., Section 3 in [1]). FlowNar operates at the frame level on unsegmented streams. It does not require ground-truth segmentation. Instead, it autonomously detects event boundaries based on $p[\texttt{SKIP}]$, allowing it to process continuous, unconstrained videos.
> > > >
> > > > [1]"Autoad: Movie description in context." Proceedings of the IEEE/CVF Conference on Computer Vision and Pattern Recognition. 2023.
> > > >
> > > > - **Offline VLMs (Qwen-VL, Videorefer, Osprey, etc.) vs. Online Narration**: We agree these models are powerful for general image/video understanding. However, they are designed for offline inference, typically receiving a complete video file as input to generate a response based on global context. FlowNar is architected for online streaming: it receives one frame at a time, updates a recurrent constant-memory state, and automatically decides *when* to narrate. Adapting these offline models to the streaming *frame-by-frame* setting is non-trivial and would require substantial architectural modifications (e.g., inclusion and fine-tuning of the special event-boundary tokens and recurrent states).
> > > >
> > > > - **Comparison Fairness and Controlled Analysis**: From an experimental design perspective, our goal is to isolate the effectiveness of the proposed streaming modules (DCM and CLAM). For this purpose, we compare against baselines (e.g., Videollm-online) that utilize the same visual and language backbones. Foundation models like Qwen-VL utilize entirely different architectures and pre-training datasets. Comparing FlowNar directly against Qwen-VL would introduce numerous confounding factors, making it difficult to determine whether performance differences stem from our streaming architecture or simply from the stronger priors of the foundation model. Therefore, we prioritize controlled comparisons against established streaming baselines to provide a clean analysis of our contributions.
> > > >
> > > > We will update our Related Work to explicitly discuss these methods, distinguishing the clip-based/audio-conditional nature of Audio Description and the offline nature of Foundation Models from the continuous, visually-conditional requirements of Streaming Narration.
> > > > We thank the reviewer again for bringing these relevant works to our attention, which will strengthen our related work section.

---

> > > > > ### Comment · Reviewer_VDde · 2025-11-28
> > > > >
> > > > > Thank you for the detailed rebuttal. These clarifications have addressed most of my previous concerns. Therefore, I maintain my recommendation for acceptance with a score of 6. If the authors open-source the code, this work will have significant impact on the Human-Object Interaction field.

---

> > > > > > ### Author Response · Authors · 2025-11-28
> > > > > >
> > > > > > We thank the reviewer for the continued support and the recommendation for acceptance. We are glad that our clarifications have helped address your concerns.
> > > > > >
> > > > > > Regarding your comment on open-sourcing: We fully agree that open-sourcing is vital for impact. We confirm that the complete training and inference code is already included in the supplementary material of this submission. Upon acceptance, we are fully committed to releasing the code, pre-trained models, and configuration files on a public GitHub repository to facilitate future research in the Human-Object Interaction and Streaming Narration fields.
> > > > > >
> > > > > > We thank you again for your time and valuable feedback which helped improve our paper.

---

### Official Review · Reviewer_YeT5 · 2025-11-01

**Soundness:** 2
**Presentation:** 2
**Contribution:** 3
**Rating:** 6
**Confidence:** 3

**Summary:**

The paper presents FLOWNAR, a comprehensive system designed for streaming long-form video narration. Its core components include Dynamic Context Management (DCM) and a newly proposed memory module, Cross Linear Attentive Memory (CLAM). Furthermore, the authors introduce a more realistic autoregressive evaluation protocol to better simulate real-world deployment scenarios. Experiments conducted on three major benchmarks — Ego4D, EgoExo4D, and EpicKitchens-100 — demonstrate significant improvements over existing streaming methods under the autoregressive setting, while also achieving substantial efficiency gains in memory usage and throughput (supporting up to 10× longer videos and approximately 3× higher FPS). These findings are well supported by quantitative results and extensive ablation studies presented in the paper.

**Strengths:**

1. This paper introduces a more rigorous and realistic autoregressive evaluation protocol, establishing a stricter yet more meaningful benchmark for this research domain.

2. In this paper, the authors address the issue of unbounded visual KV cache accumulation in existing streaming models by introducing *Dynamic Context Management (DCM)*, which enables the practical processing of arbitrarily long videos from an engineering standpoint.

3.The experiments in the paper demonstrate a remarkable improvement in efficiency, highlighting the strong practical value of the proposed approach. Moreover, the ablation studies are thorough and well-designed, clearly validating the necessity and superiority of each module, which lends strong credibility to the paper’s conclusions.

4. The work addresses a critical scalability bottleneck in streaming video models and demonstrates a feasible pathway toward real-time, long-duration video narration.

**Weaknesses:**

1.The experiments are conducted exclusively on Ego4D, EgoExo4D, EpicKitchens-100. While these benchmarks are standard for video narration, they represent a narrow domain. It remains unclear whether the proposed system generalizes effectively to more diverse visual conditions, such as outdoor scenes, dynamic camera motion, or different frame rates. The paper would benefit from broader evaluation to substantiate its claim of scalability to arbitrary long-form videos.

2.The paper briefly mentions increased training time but omits quantitative details about GPU hours, memory usage, or batch configurations. Without such data, it is difficult to assess the method’s scalability and resource efficiency in real-world scenarios. Full reproducibility would require more transparent reporting of hyperparameters, implementation details, and training cost metrics.

3. The paper lacks an in-depth analysis of the relationship between video length and the performance of CLAM. Since CLAM compresses variable-length visual histories into a fixed-size representation, it remains unclear whether this process leads to significant information loss. The discussion of CLAM’s potential failure cases is insufficient, and the newly introduced triggering mechanism lacks a detailed sensitivity analysis of its parameters.

**Questions:**

1. The experiments focus exclusively on Ego4D, EgoExo4D, and EpicKitchens-100, which primarily involve egocentric indoor scenes. Could the authors provide additional evidence or discussion regarding the generalization of FLOWNAR to more diverse video domains—such as outdoor environments, dynamic camera movements, or variable frame rates? If conducting new experiments is infeasible, a qualitative analysis or case study on different video distributions would help clarify the scalability claim.

2. The paper briefly mentions increased training time but does not provide quantitative details. Could the authors include a summary table reporting the GPU type, total training hours, peak memory usage, batch size, and learning rate configuration? Such information is crucial for assessing the practical scalability and reproducibility of the proposed system. Additionally, have the authors considered releasing training logs or scripts to ensure transparency and facilitate community validation?

3. Since CLAM compresses variable-length visual histories into a fixed-size representation, how does its performance scale with increasing video length? Is there a point where compression leads to noticeable degradation in temporal or semantic fidelity? It would also be valuable to understand any observed failure cases where CLAM struggles to maintain long-range consistency. Furthermore, could the authors provide a sensitivity analysis for the parameters of the new triggering mechanism (e.g., thresholds or update intervals) to better illustrate its stability and robustness?

---

> ### Author Response · Authors · 2025-11-22
>
> We thank the reviewer for highlighting the "remarkable improvement in efficiency" and "strong practical value" of FlowNar. We appreciate your thorough review and the suggestion to broaden the evaluation, which we believe has strengthened the paper.
>
> 1. **Narrow video domains**:  We would like to clarify that our scalability claim refers specifically to the ability of the system to process videos of arbitrary length under streaming constraints, rather than generalization across all possible visual domains. Consistent with prior online narration baselines (Videollm-online, Videollm-mod, LION-FS), we perform experiments on Ego4D and EgoExo4D, and we further extend the evaluation to EpicKitchens100. While EpicKitchens100 and EgoExo4D are primarily indoor datasets, Ego4D includes both indoor ($\sim$3100h) and outdoor ($\sim$780h) scenes. Moreover, as all three datasets are egocentric. The used datasets thus already contain outdoor environments and dynamic camera movements. For a fair comparison with prior work, we kept the frame-rate at 2 FPS. As mentioned in the response to Reviewer vKgD, changing the frame-rate requires additional training on videos with a different frame-rate. We furthermore show several qualitative examples where we apply our Ego4D-trained model directly on ActivityNet without any finetuning (Fig. 6 and Fig. 13 in the revision). Despite the domain gap, FlowNar is able to correctly identify key human activities, such as painting and playing a game.
>
> 2. **Hyperparameters, implementation details, training time**: We would like to clarify that implementation details such as GPU type, batch size, learning rate, and other hyperparameters are already provided in Section A.2.6 of the supplementary material. The total training time of the 1B-parameter model on Ego4D is 67 GPU-hours on H100s, with a peak memory usage of 47 GB. Videollm-online requires for the same hardware 36 GPU-hours of training. The cache memory for inference is reported in Table 1, and the total GPU memory usage for inference is shown in Figure 1. Regarding reproducibility, we have already provided our full training and inference code in the supplementary material, and we will release pretrained model checkpoints.
>
> 3. **Relationship between video length and performance**: To analyze the relationship between video length and CLAM’s performance while avoiding confounding factors such as error accumulation, we conducted an experiment on EpicKitchens100 under the oracle protocol, where CLAM receives ground-truth history. We group videos into four duration ranges and report performance for each group:
>
> | Duration (min) | PPL ↓ | TimeDiff ↓ | Fluency ↑ | LM-Corr. ↑ |
> | -------------- | ----- | ---------- | --------- | ---------- |
> | ≤ 2            | 2.254 | 1.749      | 39.0%     | 52.6%      |
> | 2–4            | 2.337 | 3.057      | 43.1%     | 50.1%      |
> | 4–8            | 2.188 | 3.181      | 42.0%     | 52.7%      |
> | ≥ 8            | 2.279 | 3.326      | 42.0%     | 52.4%      |
>
> The results in the table show no notable degradation as video length increases when CLAM operates with ground-truth history. This suggests that, in practice, the fixed-size state does not become a bottleneck for long-form narration when accurate historical context is provided. We agree that a fixed-size state necessarily compresses long-range information. However, narration primarily relies on fine-grained details for the current segment and more abstract contextual cues for distant history, rather than frame-level precision over the entire past. CLAM is explicitly designed to preserve the latter, while detailed frame information is retained within each current segment. This division of roles has proven effective in our experiments.
>
> 4. **Failure cases**: We have added a qualitative failure analysis in Sec. A.4 (Fig. 14) in the revised paper. We observe that errors typically manifest as fine-grained object misclassifications (e.g., predicting contextually plausible objects like a "knife" instead of the active "spatula"). We attribute this to the spatial compression ($1\text{ CLS} + 3\times3$ pooled tokens), consistent with prior streaming baselines (e.g., Videollm-online), which is required to maintain high streaming throughput. While this design ensures real-time performance, it can oversmooth the visual details necessary for precise object grounding. We also observe that severe visual degradations, such as extreme low light, can occasionally cause missed triggers.

---

> > ### Author Response · Authors · 2025-11-22
> >
> > 5. **Sensitivity analysis of triggering mechanism**: Regarding the triggering mechanism, we already provide threshold ablations in Table 11 of the supplementary material. To further complement this, we additionally include an ablation on the refractory period:
> >
> > | Ref. Period (s) | Prec. ↑   | Rec. ↑    | F1 ↑      | C ↑       | M ↑       | R ↑       |
> > | --------------- | --------- | --------- | --------- | --------- | --------- | --------- |
> > | 2               | 36.79     | **41.67** | **37.19** | 44.27     | 14.92     | 45.66     |
> > | 4               | **49.12** | 23.12     | 29.12     | **46.63** | **14.93** | **46.25** |
> > | 8               | 42.37     | 9.63      | 14.28     | 35.17     | 13.99     | 45.63     |
> >
> > When the refractory period is larger than the average segment duration (4s), the metrics drop. This is expected since the narration frequency is then forced to be lower than in the ground-truth.

---

### Official Review · Reviewer_vKgD · 2025-11-01

**Soundness:** 3
**Presentation:** 3
**Contribution:** 2
**Rating:** 6
**Confidence:** 4

**Summary:**

This paper proposes a vision-language framework to support real-time scalable video narration. It addresses the main limitation of prior streaming based vision-language model, e.g. videollm-online whose memory and computation costs grow linearly with the video length. In this paper, the authors proposed two main architecture changes to address this:
* It introduced a cross linear attentive memory mechanism that reformulates the linear attention into a sequential compressor. It can uses fix number of tokens that attend to new observations and gets updated with time on the fly.
* It uses dynamic context management strategy that prunes the past visual and text caches after each narration segment. This prevents memory built up and make the system can handle long-context windows.

The paper performs extensive ablation studies to show the effectiveness of the two proposed modules on Ego4D, EgoExo4D and EpicKitchen datasets. It can achieve 10x longer videos and 3x higher throughput than videollm baselines. In addition, the authors propose an autoregressive protocol that evaluate the full narration autoregressively on long context. In this evaluation, the method outperform previous methods reasonable well (except in the MAC metric)

**Strengths:**

Overall the method is well motivated and technically sound.

The proposed Cross Linear Attentive Memory and Dynamic Context Management can compound to each other and summarize the context on the fly with a fixed size latent space. The evaluation also demonstrates its ability to achieve better frame alignment and narration quality.

On the evaluation of narration, the method uses a combination of autoregressive long context generation and oracle evaluation. I like the autoregressive evaluation, which is a good indication of narration quality at inference time.

**Weaknesses:**

One main question (or weakness I hypothesis) as the limitation for the proposed mechanism is that the proposed summarization and pruning strategy will lead to loss of historical information, which may prevent it being used in broader tasks other than narration. The task of video narration can benefit from better historical information but may not necessarily need them in fine-grain details. Currently there is very little information about this.

My other hypothesis is that the narration may benefit from faster FPS (which is actually a strength) sampling at runtime, in particular for temporal alignment. I did not see corresponding experiments demonstrating how higher FPS will affect the result in some way.

**Questions:**

The two parts I raised in weakness are primarily the two areas I'd like to get clarity from the authors.

I am not sure whether the authors will intend to release the model for future reference. It will be good to get a confirmation.

---

> ### Author Response · Authors · 2025-11-22
>
> We thank the reviewer for assessing our method as “well motivated and technically sound” and for highlighting the effectiveness of the proposed CLAM and DCM modules. We appreciate your constructive feedback and address your specific questions below.
>
> 1. **Loss of historical information**: To support inference on arbitrarily long videos, CLAM uses a fixed-size state to summarize history. We agree that such a design necessarily compresses historical information when the temporal horizon becomes very long. However, for narration, the model primarily requires fine-grained details of the current video segment and more abstract information for distant history, rather than frame-level precision over the entire past. Our framework therefore retains detailed frame features within each current segment, while CLAM focuses on capturing longer-range, coarser historical structure. Empirically, this complementary design works well for narration, as shown in our experiments.
> For other streaming video tasks, which are not the scope of the paper, we believe that CLAM-style compression for longer-range context can be efficient as well. Note that CLAM learns what to store depending on the task, which is video narration in our case.
>
> 2. **Higher FPS**: To ensure fair comparison with prior online narration baselines (Videollm-online, Videollm-mod, LION-FS), we follow the standard 2 FPS setting used consistently in these works. Increasing FPS would indeed provide more fine-grained temporal cues, but it also substantially increases the number of tokens per video, which leads to much higher training cost. If we apply our model trained on 2 FPS directly to 4 FPS without any additional training, we observe a performance drop on EpicKitchens100 under the autoregressive protocol:
>
> | FPS   | Prec. ↑ | Rec. ↑ | F1 ↑  | C ↑   | M ↑   | R ↑   |
> | ----- | ------- | ------ | ----- | ----- | ----- | ----- |
> | **2** | 49.12   | 23.12  | 29.12 | 46.63 | 14.93 | 46.25 |
> | **4** | 48.80   | 21.35  | 26.55 | 41.77 | 14.42 | 45.41 |
>
> 3. **Release the model for future reference**: The full training and inference code is already included in the supplementary material. We will release the source code on a public GitHub repository and the pretrained models to support future research.

---

### Author Response · Authors · 2025-12-03
**Summary of the Rebuttal Phase**

Dear Area Chair,

We thank you and the reviewers for the time and effort dedicated to reviewing our work. We are pleased that the reviewers have acknowledged the novelty and efficiency of the FlowNar framework. In particular, the reviewers highlighted:


- **Significance & Efficiency:** The reviewers strongly validated the paper's core contribution to scalability, highlighting the "remarkable improvement in efficiency" and the "strong practical value" of the proposed approach. They emphasized that the combination of our modules provides a "complete solution to the scalability challenge," addressing the critical bottleneck where existing methods grow linearly with video length.

- **Methodological Soundness:** The reviewers commended the architectural design, assessing the method as "well motivated and technically sound". In particular, the CLAM module was noted as an "interesting adaptation of linear attention mechanisms" that successfully maintains constant memory and computation.

- **Rigorous Evaluation Protocol:** The reviewers acknowledged the importance of our realistic evaluation setup. They noted that our autoregressive protocol establishes a "stricter yet more meaningful benchmark" for the domain, and praised the "first-align-then-evaluate" procedure as a "thoughtful solution" to the challenge of evaluating unaligned streaming predictions.

During the rebuttal period, we actively engaged with the reviewers to address their remaining questions, which primarily focused on robustness analysis across broader conditions (e.g., long-term drift and domain generalization).
To address these points, we integrated comprehensive additional analyses into the revision. Specifically, we added a rigorous drift analysis on EpicKitchens-100 (Appendix A.3.1),
demonstrating the robust performance of FlowNar w.r.t. video duration when accurate history is provided (Table 16), and quantifying the expected autoregressive drift caused by the accumulation of feedback errors (Table 17).
Additionally, we added zero-shot qualitative results on ActivityNet (Sec.4.4 and Appendix A.4), demonstrating that despite significant domain shift, FlowNar is able to identify activity events without fine-tuning.

Notably, following these clarifications, Reviewer VDde explicitly stated that "These clarifications have addressed most of my previous concerns" and maintained their recommendation for acceptance.
Furthermore, regarding our commitment to code release, Reviewer VDde noted: "If the authors open-source the code, this work will have significant impact on the Human-Object Interaction field."

We have confirmed that the full code is included in the supplementary material and are committed to its public release.
We believe we have successfully resolved the key questions raised during the review process. We are confident that FlowNar makes a timely contribution to scalable streaming video understanding.

Sincerely,

The Authors

---

### Note · Program_Chairs · 2026-01-17
**Submission Desk Rejected by Program Chairs**

The following references in this submission do not refer to real documents and/or have major errors in bibliographic information:

 Lei Liu, Jing Zhao, Tianyuan Liu, Wen-Chin Huang, Tom Ko, Ming Fai Chan, and Helen Meng. TOVA: Transformers require optimism for viable attention. arXiv preprint arXiv:2306.00207, 2023b.